# Structure-Guided Engineering of a Family IV Cold-Adapted Esterase Expands Its Substrate Range

**DOI:** 10.3390/ijms23094703

**Published:** 2022-04-24

**Authors:** Nehad Noby, Rachel L. Johnson, Jonathan D. Tyzack, Amira M. Embaby, Hesham Saeed, Ahmed Hussein, Sherine N. Khattab, Pierre J. Rizkallah, D. Dafydd Jones

**Affiliations:** 1Department of Biotechnology, Institute of Graduate Studies and Research, Alexandria University, Alexandria 21526, Egypt; amira.embaby@alexu.edu.eg (A.M.E.); hesham25166@alexu.edu.eg (H.S.); ahmed.hussein@alexu.edu.eg (A.H.); 2Molecular Biosciences Division, School of Biosciences, Cardiff University, Cardiff CF10 3AX, UK; johnsonrl3@cardiff.ac.uk; 3European Molecular Biology Laboratory-European Bioinformatics Institute, Wellcome Genome Campus, Hinxton CB10 1SD, UK; tyzack@ebi.ac.uk; 4Department of Chemistry, Faculty of Science, Alexandria University, Alexandria 21321, Egypt; sh.n.khattab@gmail.com; 5Cancer Nanotechnology Research Laboratory (CNRL), Faculty of Pharmacy, Alexandria University, Alexandria 21521, Egypt; 6School of Medicine, Cardiff University, Cardiff CF14 4XN, UK; rizkallahp@cardiff.ac.uk

**Keywords:** serine esterase, protein engineering, enzyme structure, protein stability, substrate specificity

## Abstract

Cold active esterases have gained great interest in several industries. The recently determined structure of a family IV cold active esterase (EstN7) from *Bacillus cohnii* strain N1 was used to expand its substrate range and to probe its commercially valuable substrates. Database mining suggested that triacetin was a potential commercially valuable substrate for EstN7, which was subsequently proved experimentally with the final product being a single isomeric product, 1,2-glyceryl diacetate. Enzyme kinetics revealed that EstN7’s activity is restricted to C2 and C4 substrates due to a plug at the end of the acyl binding pocket that blocks access to a buried water-filled cavity. Residues M187, N211 and W206 were identified as key plug forming residues. N211A stabilised EstN7 allowing incorporation of the destabilising M187A mutation. The M187A-N211A double mutant had the broadest substrate range, capable of hydrolysing a C8 substrate. W206A did not appear to have any significant effect on substrate range either alone or when combined with the double mutant. Thus, the enzyme kinetics and engineering together with a recently determined structure of EstN7 provide new insights into substrate specificity and the role of acyl binding pocket plug residues in determining family IV esterase stability and substrate range.

## 1. Introduction

The urgent need for sustainable eco-friendly chemical approaches has led to the search for alternatives to traditional catalysts [1,2]. Enzymes are one such source due to their functional plasticity, need for less harsh conditions (low temperature and pressure, low organic solvent use) and ability to be biodegraded [3,4,5]. Esterases are especially useful as they can utilise a variety of ester substrates, catalysing their hydrolysis into their constituent alcohol and acid [6,7,8]. Cold active/adapted enzymes are attracting particular attention as their optimal activity at lower temperatures can be applied to temperature-sensitive processes [9,10,11,12,13,14] such as: the pharmaceutical industry for synthesising fragile chiral compounds [15,16]; detergent industry for cold washing [17]; and environmental bioremediation [18]. Cost competitiveness is a fundamental consideration for implementing bio-catalysis in the industry. Usually, the environmental advantages alone cannot support the replacement of conventional chemical catalysts. Enzymes with high activity, required specificity and stability are key determinants to guarantee high performance and low-cost production approaches. However, the catalytic properties of native enzymes may not fully meet the industrial requirements leading to them being engineered to better fulfil a particular requirement [19].

We have recently determined the structure and dynamics of cold active esterase, termed EstN7, from *Bacillus cohnii* strain N1 [20] (Figure 1a). EstN7 belongs to family IV esterases, synonymously known as HSL (hormone-sensitive lipase) family. The active site serine of family IV esterases is part of the Gly-X-Ser-X-Gly consequence sequence, which together with histidine and aspartic acid constitute the catalytic triad common to all esterases. Moreover, the oxyanion hole is comprised of the HGGG motif found in many related esterases [21]. Unlike many esterases isolated from psychrophilic organisms that still have optimal activity at >20 °C, EstN7 is truly cold active with an optimal temperature of <20 °C [22]. Its low-temperature activity profile is coupled to relatively high thermal stability, with a melting temperature of 51 °C [20]. EstN7 also retained function in the presence of up to 30% of various organic solvents, which makes it potentially useful in various applications, such as fine chemical synthesis and pharmaceutical industries [20]. Analysis of the structure and dynamics of EstN7 suggests a water entropy-based cold adaptation mechanism rather than changes in local and global dynamics [20]; the surface of EstN7 is optimised to make interactions with water and so retains the water shell even at lower temperatures when water viscosity increases. The retention of a water shell may also explain EstN7’s tolerance to organic solvents [12].

The substrate binding site of family IV esterases is comprised of an alcohol and acyl binding pocket (Figure 1b) [6,23] with the acyl binding pocket acting as the main restriction on substrate range (Figure 1c). Family IV esterases can sample a relatively broad substrate range in terms of the length of acyl binding. For example, the heroine esterase (HerE) has a specificity for short acyl chain substrates [24] whereby the thermophilic PestE has a relatively broad substrate range capable of hydrolysing longer acyl chain substrates [25]. As part of EstN7’s cold adaptation mechanism, we speculated that improved access to the active site may play a role. The N-terminal cap (Figure 1a) is common to esterases and plays a critical role in function [7,13]; it contributes to forming the walls of the main channel by which substrate accesses the active site, which is located at the bottom of a cleft. In EstN7, the N-terminal cap forms a “bridge”-like structure which results in additional channels for the substrate to reach the active site. These additional channels are missing from the closest related mesophilic and thermophilic structural homologues [20].

Here we focus on the substrate specificity of EstN7 and how knowledge of its structure allowed us to identify potentially useful commercial substrates and guided our engineering efforts to probe the molecular determinants of substrate range. Using a bespoke version of the enzyme biotransformation data-mining software Transform-MinER, built on known esterase reactions, we identified a useful putative substrate. Using structure-guided protein engineering we also successfully expanded EstN7’s substrate range from exclusively short carbon chain substrates to hydrolyse longer-chain substrates (up to C8). Identification of a suppressor mutation (N211A) that comprised the acyl binding site proved critical to expanding substrate specificity.

## 2. Results and Discussion

### 2.1. Identification of Potential Substrates for EstN7

The inherent reaction catalysed by esterases is important for various industrial applications but knowing which reactions are feasible and the potential product range can be a major restriction. The basic physicochemical properties of EstN7 suggest that it could play a role in an industrial process, including optimal activity at 5–20 °C, thermal stability, organic solvent stability and metal ion sensitivity [20,22]. EstN7 is dimeric with an α/β fold similar to that found in other family IV esterases (Figure 1a) [6,20,26]. The catalytic triad is comprised of S157, H284 and D254 (Figure 1a), and is located at the bottom of a cleft formed by the lid (orange section in Figure 1a). An ethylene glycol moiety was found bound within the active site that may act as a substrate/product mimic given that one alcohol group is within the H-bond distance of the S157 hydroxyl group (Figure 1a).

Using the available structure of EstN7, we were able to mine databases to identify potentially useful substrates for the esterase. Ideally, a commercially attractive esterase will be able to catalyse turnover of a range of substrates thus introducing the idea of functional plasticity or being catalytically precise so generating a desired single product [13,27]; the latter is demonstrated here with the synthetic substrate triacetin being hydrolysed at only a single terminal acetyl unit (Figure 2a and Appendix A). Virtual screening based on the EstN7 adapted version of Transform-MinER [28] identified a number of reactions with high similarity to known reactions with potential commercial value based on recent purchase prices. One reaction identified with potential commercial value was hydrolysis of triacetin (glyceryl triacetate) at the terminal acetyl position to form 1,2-glyceryl diacetate (1-(acetyloxy)-3-hydroxypropan-2-yl acetate). Triacetin is the simplest of all the triacylglycerides (Figure 2a) and is itself an important commercial compound with its uses in the pharmaceutical industry as a biodegradable gel system for drug delivery [29]. Glyceryl diacetate, normally in the form of an isomeric mixture of the 1,2 and 1,3 forms (Figure 2a), has numerous uses in the industrial and consumer sectors, including the food industry. In our analysis of recent purchase prices, the terminal mono-deacetylated product is highlighted as more valuable: for instance, a supply of 10 g of the terminal mono-deacetylated product is ~US$ 3000 compared to 10 g of triacetin for USD 27.

There are three potential hydrolysable ester bonds in triacetin (Figure 2a). Substrate docking of triacetin to EstN7 suggested that one of the terminal ester linkages would be the primary target for hydrolysis (Figure 2b). Analysis of the reaction products reveals that only one acetyl group is liberated as the endpoint of the EstN7 catalysed reaction, with NMR analysis of the reaction products suggesting either hydrolysis of the terminal 1 or 3 positions but not in the central C2 carbon position (Appendix A and accompanying analysis). Thus, EstN7 can potentially be used to convert triacetin into the commercially valuable and isomerically pure 1,2-glceryl diacetate.

### 2.2. EstN7^WT^ Is Restricted to Short Acyl Chain Length Substrates

The natural substrate of EstN7 is currently unknown, as is the case with many natural esterases. A common approach to determining esterase kinetics and assessing substrate range is through extending the acyl chain length of a colourimetric substrate as the acyl binding pocket is normally the main limiting factor to substrate range (Figure 1b,c) [6,23]. Enzyme kinetics with a range of *p*-nitrophenyl (*p*-NP) substrates with different length carbon chains (see Figure 1b for the scheme) shows that wild-type EstN7 prefers short-chain substrates (Table 1). *p*-NP acetate (C2) is by far the preferred substrate with a turnover of *p*-NP butyrate (C4) ~3 orders of magnitude less efficient. There was no detectable activity for longer-chain substrates *p*-NP hexanoate (C6) and *p*-NP octanoate (C8). Comparing the C2 and C4 kinetics, the major change was in *k*_cat_ (~2300-fold decrease) while *K*_M_ remained similar. This suggests that C2 and C4 bind with similar affinity but that C2 is bound in such a manner that it is more likely to undergo catalysis.

Analysis of the EstN7 structure and substrate docking provides a rationale for the restricted substrate range to short-chain units. The C2 substrate binds to place the hydrolysable ester bond close to Ser157 (Figure 3a). The alcohol binding site is open and directly faces the channel opening. The EG^1^ molecule in the active site (Figure 1a) essentially binds within the alcohol binding pocket, in keeping with its chemical character. The acyl binding pocket is buried further into the protein and has a much more restricted volume. It is the acyl binding pocket that restricts the activity of EstN7 to the short-chain substrates used here. Residue M187 together with N211 form a plug at the base of the acyl binding pocket so blocking access of longer-chained substrates to a water-filled internal cavity, with M187 providing most of the steric blocking (Figure 3a,b and Appendix A).

Comparison of EstN7 with the closest structural relatives provides further insight into the substrate specificity around the plug (Figure 3a,c and Appendix A). The plug is common to some close relatives of EstN7 such as LipW [31] and HerE [32] (Appendix A). The closest relative, the putative heroin esterase HerE (PDB 1LZK) [32] has an acidic plug due to the replacement of methionine (M187) with a glutamate (E190) (Appendix A). The asparagine (N211) in EstN7 is replaced by an alanine (A214) but this is not enough to fully open the HerE plug. LipW (PDB 3QH4) [31] has two shorter residues in place of M187 and N211 (V192 and A215, respectively) but the plug is completed by M218 that positions the sulphur atom at a position similar to that of M187 (Appendix A). PestE (PDB 2YH2) [30] does not have the plug (Figure 3c). N211 is replaced by a methionine (M214) but the backbone trajectory is away from the plug. M187 is replaced with alanine (A187) and the W206 with a leucine (L209). These sequence and conformational changes remove the plug and are the likely reason why PestE can hydrolyse longer-chain substrates [25].

### 2.3. Probing the Molecular Determinants of EstN7 Substrate Range by Structure Guided Engineering

There has been a lot of focus on identifying natural esterases with the required activity [33,34] or engineering existing esterases to widen their substrate range [35,36]. Various studies have shown that altering the residues of the acyl binding tunnel can alter chain length specificity [37,38]. Based on analysis of the EstN7 structure, three residues that comprise the acyl binding pocket plug were targeted for mutagenesis: M187, N211 and W206 (Figure 3a,b). As mentioned above, a water-filled cavity beyond the plug is also observed in related esterase structures but the plug is absent in PestE, which is likely to be the main reason this esterase can hydrolyse longer substrates (Figure 3b) [30].

In silico modelling of the M187A mutation suggests that the plug will open so allowing substrates up to C8 to bind (Figure 3d and Appendix A). However, the introduction of the M187A mutation had a major disruptive effect on the folding and stability as EstN7^M187A^ was produced in *E. coli* as inclusion bodies. Rigidity analysis undertaken previously [20] revealed that M187 forms four hydrophobic interactions, indicating that it is involved in the network of hydrophobic interactions stabilising the protein’s core structure. The loss of interactions on mutating M187 to alanine may account for its destabilising effect.

We next targeted N211A. While modelling suggested that N211A would not totally remove the plug but does not form any side-chain interactions with other residues. This potentially leaves N211 “free to mutate”. The N211A variant protein was produced as soluble and functional, with the most notable was the effect on stability. Circular dichroism (CD) spectroscopy shows that EstN7^N211A^ had a higher proportion of helical character than the wild-type at 20 °C suggesting a higher degree of secondary structure content for the mutant at this temperature (Figure 4a). Thermal unfolding of EstN7^N211A^ monitored by CD at 222 nm revealed a major transition between 55 °C and 65 °C, with a T_m_ of 60 °C (Figure 4b). As we reported previously [20], EstN7^WT^ has a more extended transition (45–60 °C) with a shallower slope and a T_M_ of 51 °C. Both EstN7^WT^ and EstN7^N211A^ had similar spectra after 70 °C suggesting that they had reached the same structural endpoint (i.e., denatured) at the end of temperature ramping (Appendix A). While is it not clear how N211A stabilises EstN7 it is present in the first turn of helix 7; asparagine has a relatively low helical propensity so switching to alanine may impact the stability of helix 7 and thus the protein as a whole.

The N211A mutation also improved catalytic efficiency towards C4 substrate by 2.4 fold compared to EstN7^WT^, with *k*_cat_ increasing 30 fold. The consequence was a drop in activity towards C2, with EstN7^N221A^ retaining ~2% of the catalytic efficiency of EstN7^WT^; the biggest contributor was a 15-fold increase in *K*_M_. The N211A mutation opened up the ability to hydrolyse C6 but not C8 substrate (Table 1).

As structural analysis suggested that M187 plays the main plug forming role (Figure 3a), the stabilising N211A mutation was combined with M187A mutation to generate a double mutant. The M187A-N221A double mutant (EstN7^DM^) protein was produced as a soluble and active protein, with the CD spectrum having a similar profile to EstN7^WT^ albeit with a shallower through at 222 nm (Figure 4a). This suggests that the stabilising effect of N211A offset the disruptive nature of the M187A mutation. EstN7^DM^ had observable activity on all tested substrates (C2 to C8) demonstrating that the mutations had expanded the substrate range of the esterase (Table 1). Activity towards C2 is still significant but is ~2% that of the wild-type enzyme and is now in a comparable range to C4 and C6 substrates in terms of catalytic efficiency. Activity towards C4 is nearly an order of magnitude higher compared to EstN7^WT^, with *k*_cat_ just over 50-fold higher. EstN7^DM^ activity towards C6 is also improved compared to EstN7^N211A^, with *k*_cat_ increasing almost 10-fold. EstN7^DM^ can now catalyse the hydrolysis of C8 substrates, albeit poorly. Indeed, EstN7^DM^ can be considered a more generalist enzyme due to its ability to hydrolyse substrates of different chain lengths. The role of stabilising or suppressor mutations allowing functionally useful but structurally deleterious mutations to be productively expressed in a particular protein is important to the natural and directed evolutionary process [39,40,41]. Here, we inadvertently discovered a suppressor mutation [N211A] in EstN7 but it clearly shows the impact such mutations can have.

W206 appears to be important for EstN7 function as mutation to alanine had a major effect on both *k*_cat_ and *K*_M_ with catalytic efficiency for C4 lower than that observed for EstN7^WT^ (Table 1). W206A appeared to shift substrate specificity towards longer-chained substrates with a 5.4 and 2.7 fold increase in catalytic efficiency to C4 and C6 compared to C2 as a substrate. Incorporating W206A into the EstN7^DM^ to create the triple mutant had little overall effect, with catalytic efficiency towards C2, C4 and C6 dropping 5, 85 and 10,000 fold, respectively, compared to the double mutant (Table 1); the triple mutant did display the highest catalytic efficiency towards C8 but only slightly and was largely similar to EstN7^DM^. While W206 can vary between leucine and phenylalanine in its nearest structural homologues (Figure 3c and Appendix A), a large hydrophobic amino may be a requisite rather than a small side-chain amino acid; W206 is an important hydrophobic tethering residue with one face of the indole ring available to potentially interact with the substrate. It could also impose a geometrical restraint on the substrate size and route for the correct approach into the active site, blocking off non-productive access.

## 3. Materials and Methods

### 3.1. Structural Analysis

The structure of EstN7^WT^ was determined previously [20]. Docking the *p*-NP-C2 and *p*-NP-C8 substrates (Sigma, Ronkonkoma, NY, USA) with the enzyme was performed manually using previous knowledge of esterase structures and the position of the ethylene glycol moiety located in the active site of the present structure to guide the process (see Figure 1a). K4V, the PDB entity defining the C2 substrate, was used, which was manipulated in COOT [42] and placed at geometrically acceptable distances from the surrounding active site lining. This coarse positioning was refined with REFMAC [43] to regularise the geometry of the model. Three different poses were tried, yielding a preferred result, based on best overlap with the ethylene glycol molecule and charge complementarity with the surrounding side chains. When the preferred docking configuration was extended to the C8 substrate version (constructed with JLIGAND [44]), in conjunction with targeted mutations introduced in silico, it proved to be the putatively correct orientation upon geometry regularisation with REFMAC as judged by the lack of steric repulsion by the non-polar side chains, and deeper insertion of the extended acyl group into the pocket by around 3.0 Å. Energy interactions resulting from the docking of the putative substrates were calculated with PISA, yielding 2.3 kcal/mol. The entropic effect of replacing the waters in the enclosed vestibule with an aliphatic side chain could not be estimated. The lack of higher energy interaction would be consistent with a wide vestibule able to accommodate a variety of entities without clashes. Thermal stability was determined by circular dichroism (CD) spectroscopy Mas described previously [20].

### 3.2. Protein Engineering and Protein Production

EstN7 mutants were generated by inverse PCR using the plasmid encoding the EstN7^WT^ gene as a template. The primers pairs used to introduce each mutation were: M187A (Forward 5′-CCAgctATTGATGATAAAAACAATTCACC-3′; Reverse 5′- ATATAATGGCATTTGGAAGCAAAG-3′), N211A (Forward 5′-TCATGATTTAgctGAAAAAGGTTGGTCTAT-3′; Reverse 5′-TTCCAGATTAGATTGCCTGTAATCTC-3′) and W206A (Forward 5′-CTAATCgcGAATCATGATTTAAACG; Reverse 5′- ATTGCCTGTAATCTCTAAGCTG). After verification of the mutations by DNA sequencing, the mutant constructs were used to transform *E. coli* BL21 DE3 cells and the variants were produced and purified as previously [20].

### 3.3. Enzyme Activity and Kinetics

The protein concentration of purified enzymes was determined using the Bradford assay [45]. The hydrolytic activity of EstN7 and its variants was determined spectrophotometrically using *p*-NP-derivatives through measuring the liberated *para*-nitrophenol at 410 nm. The reaction was performed at 20 °C in 50 mM Tris-HCl buffer (pH 8.0) with enzyme concentration adjusted appropriately enzyme (range 0.005–14 µM) depending on the rate for a particular substrate. The substrate specificity of the wild-type enzyme and the mutants were determined using different chain lengths of *p*-NP derivatives: *p*-NP acetate (C2), *p*-NP butyrate (C4), *p*-NP hexanoate (C6) and *p*-NP octanoate (C8). Each substrate was prepared as a 10 mM stock in dimethyl sulfoxide (DMSO). The kinetic parameters, Michaelis constant (*K*_M_), maximum rate (V_max_) and turnover number (*k*_cat_) were calculated using Hyper32 (https://hyper32.software.informer.com/, accessed on 1 April 2022).

EstN7 hydrolytic activity towards triacetin as a substrate was assessed by pH endpoint titration. The reaction was performed at 20 °C in a glass vessel containing 15 mL Tris–HCl buffer (50 mM), 270 mM triacetin, and suitable enzyme concentration (0.5 mg/mL final concentration). At the end of the incubation period, the liberated free fatty acids were titrated against 0.03 M NaOH using phenolphthalein as an indicator. A control sample was tested in parallel to determine the original free fatty acid content in the substrate. The product was also confirmed by NMR as outlined in the Appendix A. (Measurements were carried out on JEOL, 500 MHZ, Tokyo, Japan).

### 3.4. Data-Mining for New Valuable Biotransformations

A bespoke version of the enzyme biotransformation data-mining software Transform-MinER [46] was created using experimentally confirmed substrates and products of reactions catalysed by EstN7. This bespoke version will be referred to as the EstN7 version of Transform-MinER. A public web server [28] based on reactions in the KEGG database [47] has been made available but a bespoke version enables targeted search against the confirmed reactions. Virtual screening of a dataset of ~400,000 commercially available molecules was performed using the EstN7 version of Transform-MinER, identifying reactions with high similarity to known EstN7 reactions. These were subsequently assessed for commercial interest by obtaining recent purchase prices for substrates and products from online sources and filtering for reactions with good margins. EstN7-triacetin docking was performed using HADDOCK 2.4 web server (https://wenmr.science.uu.nl/haddock2.4/, accessed on 30 October 2020) [48] with the coordinates of triacetin obtained from PubChem (https://pubchem.ncbi.nlm.nih.gov, accessed on 1 April 2022) with the PubChem CID 5541.

## 4. Conclusions

Cold active esterases are important for various industrial and biotechnological processes but limitations on their substrate specificity and range can be a major limiting factor. Using data-mining we find that EstN7 has a potential use hydrolysing triacetin to generate a defined single terminally deacylated product. There is an argument that cold-adapted esterases may be less tolerant to protein engineering approaches to expand substrate specificity due to the perceived inherent low thermal stability and the potentially destabilising effect of opening up the plug has on esterase structure. EstN7 has relatively high thermal stability (*T*_M_ 51 °C [20]), which can be enhanced through a stabilising suppressor mutation (N211A). This in turn allows the main plug opening mutation, M187A, to be incorporated. EstN7, in common with related esterases, also has a water-filled cavity beyond the acyl binding pocket plug. While the water-filled cavity is frequently observed in close relatives of EstN7, to our knowledge it is not frequently described or considered when engineering esterases to alter substrate specificity. Thus, the presence of the buried water molecules on the other side of the plug does raise the issue of their contribution to the structure-function relationship and how such esterases can be engineered to accept longer-chain substrates is worthy of future investigations. Reference [49] is cited in the Appendix A.

## Figures and Tables

**Figure 1 ijms-23-04703-f001:**
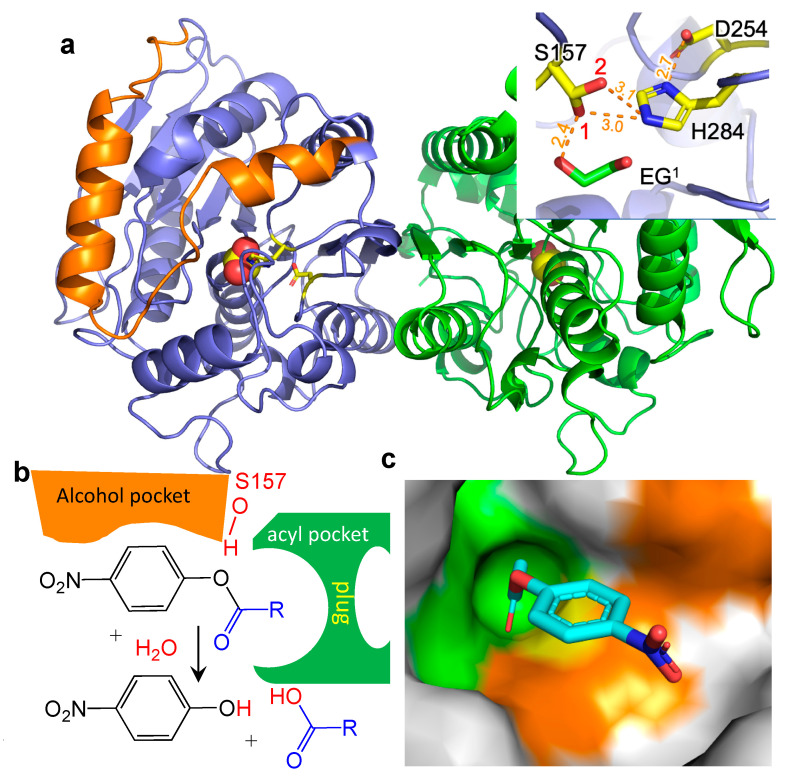
Structure of EstN7^WT^. (**a**) Overall dimeric structure of EstN7 (subunit A and B coloured blue and green, respectively) with the catalytic residues coloured yellow, while the N-terminal cap domain is coloured orange. Inset are details of the catalytic site; orange dashed lines represent the distances between individual atoms. EG^1^ is an ethylene glycol molecule found in the active site. (**b**) The esterase alcohol and acyl binding pockets, including the acyl pocket plug are shown schematically as are the products of the reaction involving the pNP substrate. (**c**) Substrate binding regions with docked *p*-NP-C2. The green region represents the acyl binding site (including the plug), orange the alcohol binding site and yellow the catalytic residues.

**Figure 2 ijms-23-04703-f002:**
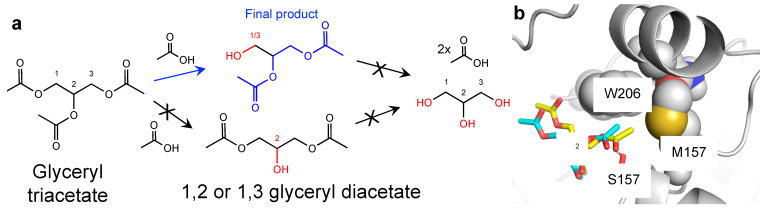
Triacetin as a substrate. (**a**) Potential routes of triacetin hydrolysis. The final observed product is shown in blue (1,2-glyceryl diacetate) with other potential products (1,3-glyceryl diacetate, the monoacetates or glycerol) not observed coloured black. (**b**) Triacetin docking to EstN7. Two docked forms of triacetin (yellow and cyan) are shown. The cluster represented the lowest HADDOCK score (−24.0 ± 0.6) observed for the simulation. The central C2 carbon of the glycerol backbone is highlighted for reference.

**Figure 3 ijms-23-04703-f003:**
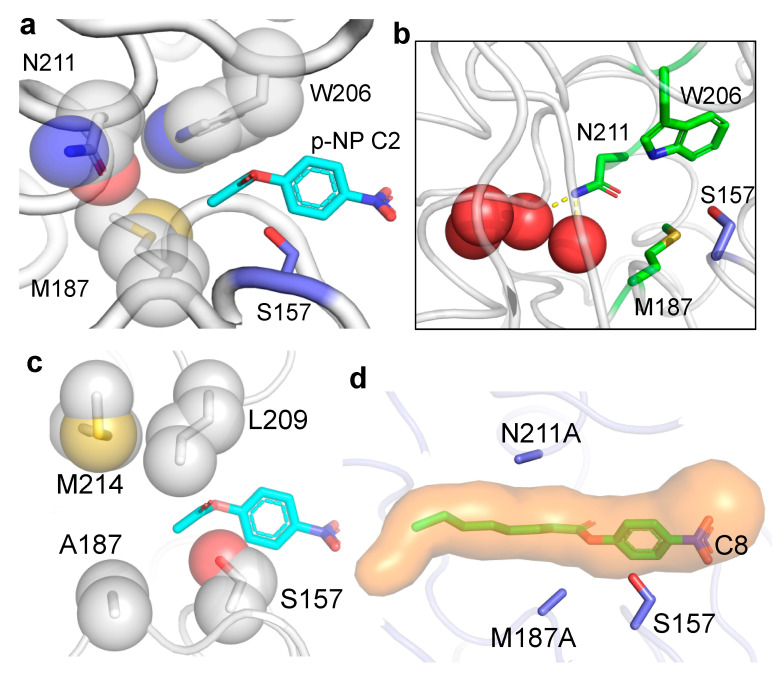
Structural basis for substrate binding and specificity in EstN7. The docked substrate *p*-NP-C2 is coloured cyan and the nucleophilic S157 is shown as blue sticks. (**a**) Docking of substrate *p*-NP-C2. The three residues contributing to the acyl binding pocket and the plug structure are shown as grey spheres. (**b**) The water (red spheres) cavity behind the plug. The plug residues M187, W206 and N211 are coloured green. Polar contacts from the carboxamide of N211 to the water molecules are shown as yellow dashes. The composition of the water cavity is detailed in Appendix A. (**c**) The open plug present in PestE [30] (PDB 2YH2). (**d**) Opening the acyl pocket plug. In silico modelling of acyl plug mutations M187A mutation N211A. The accessible tunnel is coloured orange and the *p*-NP-C8 substrate is coloured green. The model of the M187A mutation alone is shown in Appendix A.

**Figure 4 ijms-23-04703-f004:**
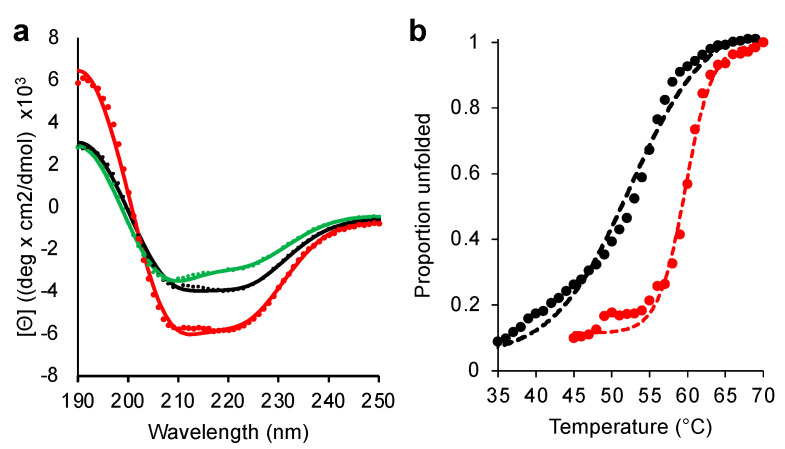
CD spectroscopy of EstN7. (**a**) CD spectra of EstN7^WT^ (black), EstN7^N211A^ (red) and EstN7^DM^ (green). CD spectra were collected at 20 °C and a Savitzky–Golay filter applied. (**b**) Temperature-dependent change in proportion of protein folded (based on molar ellipticity at 222 nm at each temperature) for EstN7^WT^ (black) and EstN7^N211A^ (red). The dashed line represents the Boltzmann sigmoidal fit of the data (circles).

**Table 1 ijms-23-04703-t001:** Substrate specificity of wild-type EstN7and its engineered variants.

EstN7 Variant	*p*-NF Substrate	*K*_M_ (mM)	*k*_cat_ (s^−1^)	Catalytic Efficiency (s^−1^ M^−1^)	Fold Change ^a^
WT	C2	0.05 ± 0.007	760 ± 3.5	1.52 × 10^7^	1
	C4	0.057 ± 0.002	0.32 ± 0.005	5614	1
	C6	-	-	-	
	C8	-	-	-	
N211A	C2	0.75 ± 0.001	280.5 ± 0.07	3.74 × 10^5^	0.024
	C4	2 ± 0.2	15 ± 0.5	7500	1.33
	C6	0.37 ± 0.05	2.3 ± 0.2	6283	-
	C8	-	-	-	-
W206A	C2	1.3 ± 0.14	1.5 ± 0.07	1192	7.8 × 10^−5^
	C4	1.2 ± 0.55	8 ± 1.0	6666	1.2
	C6	1.1 ± 0.07	3.8 ± 0.07	3347	-
	C8	-		-	-
DM (M187A/N211A)	C2	1.1 ± 0.07	361 ± 0.14	3.28 × 10^5^	0.021
	C4	0.37 ± 0.08	17.55 ± 2.1	4.68 × 10^4^	8.0
	C6	1.2 ± 0.01	24.5 ± 0.7	2.04 × 10^4^	-
	C8	0.95 ± 0.02	0.45 ± 0.02	473	-
TM (M187A/W206A/N211A)	C2	2 ± 0.3	0.07 ± 0.01	35	2.31 × 10^−6^
	C4	2.1 ± 0.4	1 ± 0.11	467	0.08
	C6	0.59 ± 0.03	1.45 ± 0.02	2457	-
	C8	2 ± 0.03	0.85 ± 0.007	425	-

^a^ compared to EstN7^WT^ activity on the same substrate.

## Data Availability

The structure of EstN7 has been deposited in the PDB under accession code 7b4q. Data will be made available via http://doi.org/10.17035/d.2021.0139101420 and http://doi.org/10.17035/d.2022.0171248569 (accessed on 1 April 2022).

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
