# Peer review of "Structure-Guided Engineering of a Family IV Cold-Adapted Esterase Expands Its Substrate Range"

_ijms, 2022, doi:10.3390/ijms23094703_

Round 1
Reviewer 1 Report
This manuscript reports engineering of Cold-active esterase from Bacillus cohnii (EstN7), in order to expand the substrate range to larger esters (C8). The authors showed that M187A mutation is crucial for reducing steric hindrance in acyl pocket to accept longer chain substrates, while at the same time detrimental to enzyme stability. Altogether, the paper is clearly presented and the experimental results are adequately documented and sufficient details are provided. I suggest acceptance of this article for publication in International Journal of Molecular Sciences with minor revision:
Figure 1(a): Is the N-terminal cap domain coloured orange? This should be mentioned in figure caption.
Table 1: I suggest including a general reaction scheme
References 11, 16 and 42 should be corrected
Author Response
We would like to thank reviewer (1) for the throughout revision of the manuscript.
All the corrections are marked up using track change, either in the main manuscript or in the supplementary file.
- Figure 1(a): Is the N-terminal cap domain coloured orange? This should be mentioned in figure caption.
Response
A statement is added in the figure caption (Figure 1a) pointing out the cap domain color.
- Table 1: I suggest including a general reaction scheme
Response
The authors think that the table has a lot of data to be summarized on a chart. In this case, 1 or 2 parameters would be displayed on the chart and this will not be informative.
- References 11, 16 and 42 should be corrected
Response
The mentioned references have been corrected.
Reviewer 2 Report
The manuscript “comStructure-guided engineering of a family IV cold-adapted esterase expands its substrate range” by Noby et al. describes the use of recently determined structure of family IV cold active esterase (EstN7) from Bacillus cohnii N1 to broaden its substrate range and to probe its commercially valuable substrates. Through enzyme kinetics and mutational studies, M187, N211 and W206 were identified as the essential residues forming the key plug. A double mutant, M187A-N211A showed to have a broader substrate specificity, capable of hydrolysing a C8 substrate compared to its WT counterpart. Taken together, this study has gained insights into the role of residues in the acyl binding pocket and the substrate specificity of EstN7.
Overall, the manuscript is well-written and a useful story to confirm the value of comStructure-guided engineering for expanding substrate specificity. After the following points have been addressed, I recommend this manuscript for publication in International Journal of Molecular Sciences.
(1). The authors have commented that the introduction of the M187A mutation had a major disruptive effect on the folding and stability as EstN7M187A was produced in E. coli as inclusion bodies. Rigidity analysis undertaken previously [20] revealed that M187 forms four hydrophobic interactions, indicating that it is involved in the network of hydrophobic interactions stabilising the protein’s core structure. The loss of interactions on mutating M187 to alanine may account for its destabilising effect.
It is understandable that introduction of mutation particularly in the hydrophobic core can cause issue in protein expression (in the case of M187) and in some cases, protein of interest is expressed as inclusion bodies. However, growth condition for protein expression has not being optimised for this particular construct. The condition used (1 mM IPTG, 18 h at 37 deg) can lead to high level expression of recombinant protein and can likely cause aggregation of expressed protein in inclusion bodies.
Other residues with similar hydrophobicity to methionine has not being tested such as Leu, Val, Ile.
It would be interesting to see the substrate specificity of M187A mutant and to understand whether this mutation alone hydrolyse C8.
(2). The triple mutant did display the highest catalytic efficiency towards C8 but only slightly and was largely similar to EstN7DM.
kcat values of EstN7DM and EstN7TM are in the same order of magnitude, suggesting that the difference is insignificant. Statistical analysis should be included.
(3). A multiple sequence alignment among EstN7, LipW and HerE will be useful to highlight key residues.
(4). In the supplementary, spectra in figure S1a and S1b are out of phase.
Author Response
We would like to thank Reviewer (2) for his/her careful revision and the constructive comments.
All corrections are marked up using track change, either in the main manuscript or in the supplementary file.
(1). The authors have commented that the introduction of the M187A mutation had a major disruptive effect on the folding and stability as EstN7M187A was produced in E. coli as inclusion bodies. Rigidity analysis undertaken previously [20] revealed that M187 forms four hydrophobic interactions, indicating that it is involved in the network of hydrophobic interactions stabilising the protein’s core structure. The loss of interactions on mutating M187 to alanine may account for its destabilising effect.
It is understandable that introduction of mutation particularly in the hydrophobic core can cause issue in protein expression (in the case of M187) and in some cases, protein of interest is expressed as inclusion bodies. However, growth condition for protein expression has not being optimised for this particular construct. The condition used (1 mM IPTG, 18 h at 37 deg) can lead to high level expression of recombinant protein and can likely cause aggregation of expressed protein in inclusion bodies.
Other residues with similar hydrophobicity to methionine has not being tested such as Leu, Val, Ile.
It would be interesting to see the substrate specificity of M187A mutant and to understand whether this mutation alone hydrolyse C8.
Response:
Different induction conditions of M187 mutant had been applied using different temperature range (15-25oC), lower IPTG concentration, and shorter induction time (6h-12h). However, this didn’t help in enhancing the enzyme folding of this mutant.
Mutating M187 into other hydrophobic residues (Leu, Val, Ile) will be considered in our future work
2). The triple mutant did display the highest catalytic efficiency towards C8 but only slightly and was largely similar to EstN7DM.
kcat values of EstN7DM and EstN7TM are in the same order of magnitude, suggesting that the difference is insignificant. Statistical analysis should be included.
Response:
A statistical analysis of Kcat values of the two mutants is performed and added in the supplementary file in the form of a chart with standard error bars (Figure S6). The significance was considered based on the overlap rule for standard error bars. The corresponding reference is added to Supplementary file.
(3). A multiple sequence alignment among EstN7, LipW and HerE will be useful to highlight key residues.
Response:
A Figure designated as S4 is added to the supplementary file to illustrate the multiple sequence alignment between EstN7, HerE, LipW, PestE .
(4). In the supplementary, spectra in figure S1a and S1b are out of phase.
Response:
Phase correction was performed. The chemical shift values were slightly shifted after phase correction. The chemical shift values were revised and corrected.